

# PromoterPredict: sequence-based modelling of *Escherichia coli* σ⁷⁰ promoter strength yields logarithmic dependence between promoter strength and sequence

Ramit Bharanikumar[1], Keshav Aditya R. Premkumar[2] and Ashok Palaniappan[3]

[1] Biotechnology, Sri Venkateswara College of Engineering (Autonomous), Sriperumbudur, Tamil Nadu, India
[2] Computer Science and Engineering, Sri Venkateswara College of Engineering (Autonomous), Sriperumbudur, Tamil Nadu, India
[3] Bioinformatics, School of Chemical and Biotechnology, SASTRA Deemed University, Thanjavur, Tamil Nadu, India

Corresponding author
Ashok Palaniappan,
apalania@scbt.sastra.edu

## ABSTRACT

We present PromoterPredict, a dynamic multiple regression approach to predict the strength of *Escherichia coli* promoters binding the σ⁷⁰ factor of RNA polymerase. σ⁷⁰ promoters are ubiquitously used in recombinant DNA technology, but characterizing their strength is demanding in terms of both time and money. We parsed a comprehensive database of bacterial promoters for the −35 and −10 hexamer regions of σ⁷⁰-binding promoters and used these sequences to construct the respective position weight matrices (PWM). Next we used a well-characterized set of promoters to train a multivariate linear regression model and learn the mapping between PWM scores of the −35 and −10 hexamers and the promoter strength. We found that the log of the promoter strength is significantly linearly associated with a weighted sum of the −10 and −35 sequence profile scores. We applied our model to 100 sets of 100 randomly generated promoter sequences to generate a sampling distribution of mean strengths of random promoter sequences and obtained a mean of 6E-4 ± 1E-7. Our model was further validated by cross-validation and on independent datasets of characterized promoters. PromoterPredict accepts −10 and −35 hexamer sequences and returns the predicted promoter strength. It is capable of dynamic learning from user-supplied data to refine the model construction and yield more robust estimates of promoter strength. PromoterPredict is available as both a web service (https://promoterpredict.com) and standalone tool (https://github.com/PromoterPredict). Our work presents an intuitive generalization applicable to modelling the strength of other promoter classes.

## INTRODUCTION

The primary *Escherichia coli* promoter-specificity factor and the one widely used in recombinant DNA technology is the $\sigma^{70}$ factor. Promoters recognized by $\sigma^{70}$-containing RNA polymerase are called core promoters and share the following features: two conserved hexamer sequences, separated by a non-specific spacer of ideally 17 nucleotides. The two hexamers are located ~35 and ~10 bp upstream of the transcription start site, and are called the −35 and −10 sequences, respectively (*Maquat & Reznikoff, 1978*; *Bujard, 1980*; *Paget & Helmann, 2003*; *Kadonaga, 2012*). −35 and −10 sequences matching the consensi motifs (TTGACA and TATAAT, respectively) are known as canonical hexamers (*Galas, Eggert & Waterman, 1985*; *Deuschle et al., 1986*; *Stormo, 1990*). It is known that the conserved hexamer regions are vital for recognizing and optimizing the interactions between DNA and the RNA polymerase (*Hawley & McClure, 1983*; *Knaus & Bujard, 1990*; *Hook-Barnard, Johnson & Hinton, 2006*; *Feklistov & Darst, 2011*; *Basu et al., 2014*).

Theory has yielded a linear relationship between the total promoter score and the natural log of promoter strength (*Berg & Von Hippel, 1987*; *Li & Zhang, 2014*). Nucleotide occurrence frequencies were first used by *Weller & Recknagel (1994)* in promoter strength prediction. Additivity in promoter-polymerase interaction has been affirmed by *Benos, Bulyk & Stormo (2002)*. Patterns in $\sigma^{70}$ promoters have been quantified by *Huerta & Collado-Vides (2003)*. Strength of *E. coli* $\sigma^{E}$ RNA polymerase promoters were studied by *Rhodius & Mutalik (2010)*. The complexity of *E. coli* $\sigma^{70}$ promoter sequences has been treated from an information theoretic standpoint by *Shultzaberger et al. (2007)*. More recently, an support vector machines (SVM) model has been successfully applied to predicting the strength of a mutation library of *E. coli* Trc promoter sequences (*Meng et al., 2017*). One drawback with an SVM or artificial neural networks (ANN) machine learning model is the 'black-box' approach; that is, the absence of any mechanistic insights that could be gleaned with respect to the relationship between promoter sequence and strength. Such an understanding could be vital in the prediction of promoter strengths in different contexts, as well as the forward design of promoters in finely-tuned genetic circuits (see *Endy, 2005*; *De Mey et al., 2007*; *Salis, Mirsky & Voigt, 2009*; *Li & Zhang, 2014*). Many freely available resources predict the location of promoters in a genomic sequence mainly by identifying the −10 and −35 regulatory sequences (*De Jong et al., 2012*), but very few tools are available to predict the strength of such sequences. One tool provides qualitative predictions ('strong' or not) of promoter strength based on the occurrence of a triad pattern (*Dekhtyar, Morin & Sakanyan, 2008*), and is available as a macro. Here, we present a two-step approach to the predictive modelling of the strength of $\sigma^{70}$ core promoters, and a companion web-based platform and Python standalone tool that implement our method along with the option to dynamically include user data into the prediction model. Our implementation is the first freely available tool/web-server for the quantitative prediction of promoter strength.

## METHODS

### Generative model of promoter sequences

A generative model of the −10 and −35 promoter sequences is constructed using two position weight matrices ($PWM_{-10}$ and $PWM_{-35}$) in the following manner.

A comprehensive set of σ⁷⁰-binding promoter sequences was extracted from the RegulonDB (*Gama-Castro et al., 2016*). For each promoter sequence, we extracted a −35 region of 13 nucleotides centred at −35 position, and a −10 region of 13 nucleotides centred at the −10 position, to allow for uncertainties in the precise position of occurrence of the hexamers. For each −35 region, we used FIMO (*Grant, Bailey & Noble, 2011*) to find the best match to the consensus −35 motif, and similarly for the −10 regions, to obtain a dataset of −35 and −10 hexamer sequences. This dataset was then filtered for only significant hits to the consensi motifs (*p*-value < 0.05) and the resulting dataset was used to determine the weights of each nucleotide at each position of the −35 and −10 hexamers. Nucleotide-wise counts at each position of the hexamer motifs were augmented by a pseudo-count prior to correct for *E. coli* GC content of 50.8% and the resulting frequency matrices were converted into log-odds matrices. Biopython routines (www.biopython.org) were used.

## Linear modelling of promoter strength

Following *Berg & Von Hippel (1987)*, we modelled the relationship between the promoter sequences and the *ln* of the promoter strength using multiple linear regression. The training set of 18 promoters is drawn from the Anderson library of activator-independent plasmid *tet* promoter variants maintained at the Registry of standard biological parts (http://parts.igem.org/Promoters/Catalog/Anderson). Each promoter sequence is scored with respect to the generative models of the −10 and −35 motifs (i.e. the $PWM_{-10}$ and $PWM_{-35}$ matrices) and the two scores obtained formed the feature space of the regression modelling. The regression coefficients to be determined represent the weights of the −10 and −35 regions in the regression analysis. The Anderson library provided promoter strengths spanning two orders of magnitude and normalized in the range 0.00–1.00 with respect to the strongest (i.e. reference) promoter. It was noted that the normalisation step would not affect a linear relationship, altering only the constant of the regression. The normalised strength values were log-transformed to obtain the required response variable values. Since the *ln* function rapidly descends towards—Inf with decreasing promoter strength, we capped the infimum of promoter strength at 0.0001 prior to log-transformation. The least-squares cost function was minimized using iterative gradient descent. The model parameters were assessed using *t*-statistics, and the overall model was assessed using F-statistic and the adjusted multiple coefficient of determination given by:

$$\text{Adj. } R^2 = 1 - \left\{ \left(1 - R^2\right) * \left[ (n-1)/(n-m-1) \right] \right\} \tag{1}$$

where *m* is the number of features and *n* is the number of instances. The adjustment is a penalty for increasing model complexity.

## Model validation

The model of promoter strength was validated in three ways:

i) The model was validated using leave-one-out cross-validation (LOOCV).

ii) We generated 100 sets of 100 randomly generated promoter sequences each, using the `sample` function in Python. From the obtained sampling distribution of mean strengths of random promoter sequences, we calculated the estimate of the true mean strength of a random promoter sequence, together with its standard error.

iii) We further validated our model on independent datasets of characterized promoters available in *Davis, Rubin & Sauer (2011)*, *Dekhtyar, Morin & Sakanyan (2008)*, and *Dayton et al. (1984)*.

## RESULTS

The entire datasets of 1,004 −35 hexamers and 1,046 −10 hexamers parsed out of RegulonDB are available as Supplementary Information. The conservation profiles of the extracted −35 and −10 hexamer sequences of the promoters in the RegulonDB were visualized and shown in Fig. 1. Based on these PWMs, the site scores of each promoter sequence in the Anderson library were regressed on the corresponding *ln* of the promoter strength. A summary of this process with the training data, log-transformation of the promoter strength and predicted response values is presented in Table 1. The modelling process converged within $10^5$ iterations by tuning the gradient descent to a learning rate (α) of 0.015, and the following model was obtained:

$$ln(\text{promoter strength}) = -5.1046 + 0.4271 * (\text{PWM}_{-35}) + 0.2726 * (\text{PWM}_{-10}) \qquad (2)$$

We derived an independent solution of the multiple regression using `R` (www.r-project.org) and obtained a correlation coefficient of 0.998 between the fitted values of the two models. The interval estimates of the coefficients of the regression were computed in `R` using `confint (fit, level = 0.95)`, and obtained the following 95% confidence intervals:

```
Intercept :       (−6.4974449, −3.7118421)
PWM_35    :       (0.2445358, 0.6095848)
PWM_10    :       (0.1434939, 0.4017307)
```

The interval estimates did not include zero, and this implied that the coefficients were significant at the 0.05 level. In fact, all the three estimates were significant at a *p*-value of 1E-3. The F-statistic of the overall regression was significant at a *p*-value of 2E-4 and adj. $R^2$ was ≈0.65. The plane of best fit corresponding to the above model is visualized in Fig. 2.

The model was then cross-validated using a 18-fold LOOCV (similar to jack-knife). Cross-validation yielded a correlation coefficient of ~0.76 (Table 2). We sought to benchmark our model on a negative test set by generating random −35 and −10 hexamer sequences. To this end, we applied our model to 100 sets of 100 random promoter sequences each (available in Supplementary Information) and estimated the true mean of the sampling distribution as 0.00055. The standard error of the estimate was 1.04E-7. The low predicted strength along with the very small standard error indicated that the

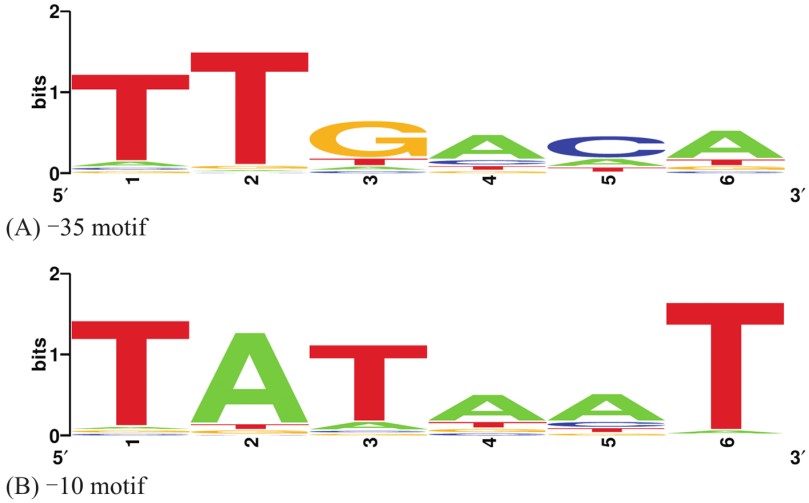

(A) −35 motif

(B) −10 motif

**Figure 1 Sequence logos of the −35 and −10 hexamers of the selected RegulonDB promoters.** (A) −35 motif; (B) −10 motif. Figure was made using WebLogo (*Crooks et al., 2004*).

**Table 1 Summary of promoter information.**

| Promoter | −35 hexamer | −10 hexamer | Promoter activity | *ln* (Promoter activity) | Predicted *ln* (Promoter activity) |
|---|---|---|---|---|---|
| BBa_J23100 | TTGACG | TACAGT | 1 | 0 | −1.6336486579 |
| BBa_J23101 | TTTACA | TATTAT | 0.7 | −0.35667494 | 0.0555718065 |
| BBa_J23102 | TTGACA | TACTGT | 0.86 | −0.15082289 | −1.0957849491 |
| BBa_J23104 | TTGACA | TATTGT | 0.72 | −0.32850407 | 0.1647181133 |
| BBa_J23105 | TTTACG | TACTAT | 0.24 | −1.42711636 | −2.2871659092 |
| BBa_J23106 | TTTACG | TATAGT | 0.47 | −0.75502258 | −1.3174788735 |
| BBa_J23107 | TTTACG | TATTAT | 0.36 | −1.02165125 | −1.0266628468 |
| BBa_J23108 | CTGACA | TATAAT | 0.51 | −0.67334455 | −0.4282477098 |
| BBa_J23109 | TTTACA | GACTGT | 0.04 | −3.21887582 | −3.3693144659 |
| BBa_J23110 | TTTAGG | TACAAT | 0.33 | −1.10866262 | −3.3946866337 |
| BBa_J23111 | TTGACG | TATAGT | 0.58 | −0.54472718 | −0.3731455955 |
| BBa_J23112 | CTGATA | GATTAT | 0.01 | −4.60517019 | −3.1533888284 |
| BBa_J23113 | CTGATG | GATTAT | 0.01 | −4.60517019 | −4.2356234817 |
| BBa_J23114 | TTTATG | TACAAT | 0.1 | −2.30258509 | −2.5943689001 |
| BBa_J23115 | TTTATA | TACAAT | 0.15 | −1.89711998 | −1.5121342469 |
| BBa_J23116 | TTGACA | GACTAT | 0.16 | −1.83258146 | −1.5897942167 |
| BBa_J23117 | TTGACA | GATTGT | 0.06 | −2.81341072 | −1.1644781255 |
| BBa_J23118 | TTGACG | TATTGT | 0.56 | −0.5798185 | −0.91751654 |

**Note:**
The promoter activities (strengths) are seen to span two orders of magnitude in the range (0.0, 1.0). The promoters follow the naming in the Anderson dataset.

model predicted these instances to be non-promoter sequences with good certainty. This affirmed the specificity of our model for true promoters.

To validate our model further on true promoter sequences and experimentally characterized promoter strengths, we used datasets available in the literature and
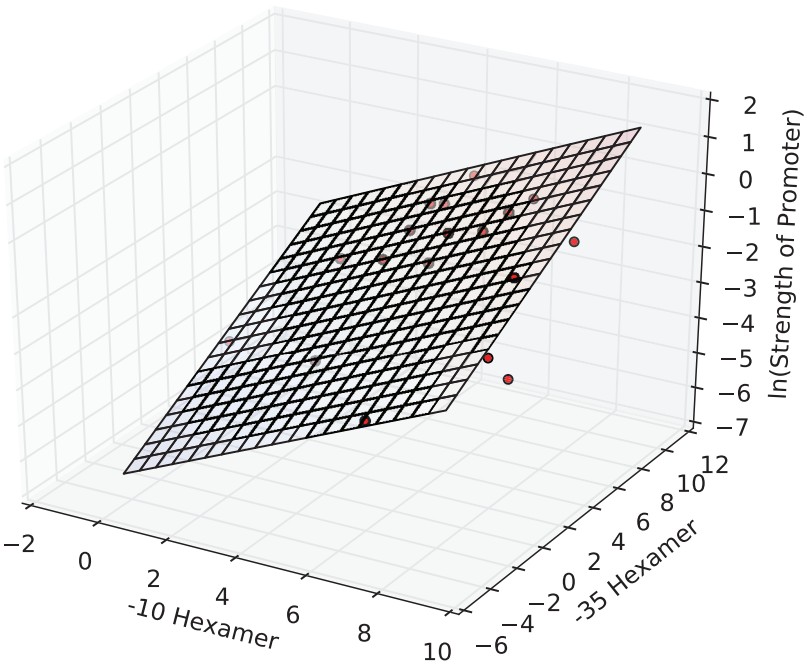

**Figure 2 The regression surface of the estimated model with the training data points (red).** *x*- and *y*-axes represent PWM scores and the *z*-axis (vertical) represents the predicted *ln*(promoter strength).

**Table 2 Cross-validation results.**

| Fold | PWM_35 | PWM_10 | Combined | logStrength | cvpred | cvres |
|------|--------|--------|----------|-------------|--------|-------|
| 1 | 6.5966 | 2.398 | 9 | 0 | −1.757 | 1.757 |
| 2 | 6.9195 | 8.089 | 15.01 | −0.357 | 0.145 | −0.50 |
| 3 | 9.1308 | 0.402 | 9.53 | −0.151 | −1.3 | 1.15 |
| 4 | 9.1308 | 5.025 | 14.16 | −0.329 | 0.286 | −0.62 |
| 5 | 4.3854 | 3.465 | 7.85 | −1.427 | −2.36 | 0.93 |
| 6 | 4.3854 | 7.022 | 11.41 | −0.755 | −1.377 | 0.62 |
| 7 | 4.3854 | 8.089 | 12.47 | −1.022 | −1.027 | 0.00 |
| 8 | 4.5119 | 10.086 | 14.6 | −0.673 | −0.362 | −0.31 |
| 9 | 6.9195 | −4.474 | 2.45 | −3.219 | −3.463 | 0.24 |
| 10 | 4.3854 | 5.462 | 9.85 | −1.109 | −1.792 | 0.68 |
| 11 | 6.5966 | 7.022 | 13.62 | −0.545 | −0.349 | −0.20 |
| 12 | 2.5179 | 3.213 | 5.73 | −4.605 | −2.847 | −1.76 |
| 13 | −0.0162 | 3.213 | 3.2 | −4.605 | −3.977 | −0.63 |
| 14 | 2.3914 | 5.462 | 7.85 | −2.303 | −2.646 | 0.34 |
| 15 | 4.9255 | 5.462 | 10.39 | −1.897 | −1.485 | −0.41 |
| 16 | 9.1308 | −1.411 | 7.72 | −1.833 | −1.518 | −0.32 |
| 17 | 9.1308 | 0.15 | 9.28 | −2.813 | −0.796 | −2.02 |
| 18 | 6.5966 | 5.025 | 11.62 | −0.58 | −0.944 | 0.36 |

**Note:**
In each fold of cross-validation, the instance corresponding to the fold was designated as the test instance while the prediction model was built using the rest of the instances. This process was repeated 18 times, once for each test instance and the cross-validation (CV) residuals were obtained. combined, sum of the PWM scores; cvpred, predicted log strength of the test instance; cvres, cross-validation residual.

compared the predicted strength with the experimental results and examined their concordance. The following results were obtained:

i) For the 10 promoters discussed by *Davis, Rubin & Sauer (2011)*, we ranked the promoters in Table 1 of the same reference according to their strengths and observed a 1,000-fold span of promoter strengths, 1E-3 to 1 (Table 3). Promoters 2 and 3 were identically strong, hence we took the average of their predicted strengths in ranking the promoters. With this arrangement, we found that the predicted order of promoters in terms of strength exactly reproduced the experimentally characterized order. Despite the fact that Anderson library and these promoters were characterized and normalized using different systems, the model was able to predict surprisingly well across a promoter strength spectrum spanning three orders of magnitude.

ii) Next, we applied our model to the set of 13 strong promoter candidates of *Thermotoga maritima* discussed in *Dekhtyar, Morin & Sakanyan (2008)*. Using the hexamer sequences provided in Fig. 5 of the same reference, we applied our model and obtained quantitative predictions of promoter strengths (Table 4). Almost all the promoters had predicted strengths >0.38 and promoters with canonical hexamers even had strengths >1.00. One promoter (TM0032) was predicted as 'weak' with a strength ~0.056 and seemed to point to an apparent anomaly in the relationship between promoter sequence and strength, possibly highlighting the need for further experimentation on this promoter. Our observations were corroborated by Fig. 4 in the same reference that showed the least and greatly reduced expression from this particular promoter. These results taken in conjunction with the results on random promoter sequences affirmed the ability of our model to discriminate between promoters at opposite ends of the strength spectrum.

iii) We also applied our model on the five promoters discussed in *Dayton et al. (1984)*. Of these, the first three are known as 'major' promoters that are active even at low concentrations of the polymerase, whereas the last two are 'minor', less strong promoters that are only active when the polymerase is present at high concentrations. We applied our model on the promoter sequences found in Fig. 5 of the same reference and found the predictions in line with the nature of these promoters (Table 5). The activity of the least strong 'major' promoter is about two times more than the activity of the strongest 'minor' promoter. Hence, our modelling approach was able to discriminate between major and minor promoters.

## DISCUSSION

In addition to the independent contributions of −35 and −10 sites to promoter strength, we were interested in exploring if any interactions between them could contribute to promoter strength. To this end, we examined the following model in R:

```
lm(logStrength ~ PWM35 * PWM10)
```

where PWM35 and PWM10 represent the corresponding site scores. This model resulted in a lower adj. $R^2$-value than that without any interactions. Further, the $p$-value of the

**Table 3 Validation results: using data of *Davis, Rubin & Sauer (2011)*.**

| Actual rank | Promoter | −35 sequence | −10 sequence | Strength | Predicted exp(logStrength) | Predicted rank |
|---|---|---|---|---|---|---|
| 1 | pro1 | tttacg | gtatct | 0.009 | 0.0079073845 | 1 |
| 2.5 | pro2 | gcggtg | tataat | 0.017 | 0.0306978849 | 2.5 |
| 2.5 | pro3 | ttgacg | gaggat | 0.017 | 0.0306978849 | 2.5 |
| 4 | proA | tttacg | taggct | 0.03 | 0.0482647297 | 4 |
| 5 | pro4 | tttacg | gatgat | 0.033 | 0.0809816409 | 5 |
| 6 | pro5 | tttacg | taggat | 0.05 | 0.0867400443 | 6 |
| 7 | proB | tttacg | taatat | 0.119 | 0.1534857959 | 7 |
| 8 | pro6 | tttacg | taaaat | 0.193 | 0.2645364297 | 8 |
| 9 | proC | tttacg | tatgat | 0.278 | 0.3059490889 | 9 |
| 10 | proD | tttacg | tataat | 1 | 0.6173668247 | 10 |

**Note:**
The promoters were ordered based on the rank of their strength, and given as input to our model. The predicted promoter log strengths were then examined for agreement with the actual rank and the ordering obtained matched the original ordering. The individual predicted values for pro2 and pro3 were 0.0024 and 0.059, respectively.

**Table 4 Validation with *T. maritima* strong promoter candidates.**

| Promoter | −35 sequence | −10 sequence | Strength | Predicted exp(logStrength) | Predicted class |
|---|---|---|---|---|---|
| TM0373 | ttgaca | tataat | Strong | 4.6845788997 | Strong |
| TM1016 | ttgaat | tttaat | Strong | 0.3808572257 | Strong |
| TM1272 | ttgaca | tttaat | Strong | 1.6386551999 | Strong |
| TM1429 | ttgaca | tataat | Strong | 4.6845788997 | Strong |
| TM1667 | ttgaaa | tataat | Strong | 2.5859432664 | Strong |
| TM1780 | ttcata | tataat | Strong | 0.463878289 | Strong |
| Tmt11 | ttgaat | taaaat | Strong | 0.4665383797 | Strong |
| TM0032 | tcgaaa | cataat | Strong | 0.0562167049 | *Weak* |
| TM0477 | ttgaat | tataat | Strong | 1.0887926414 | Strong |
| TM1067 | ttgacc | tattat | Strong | 0.7046782664 | Strong |
| TM1271 | ttgaca | tataat | Strong | 4.6845788997 | Strong |
| Tmt45 | ttgaac | tataat | Strong | 0.670434893 | Strong |
| TM1490 | ttgact | taaaat | Strong | 0.8451600149 | Strong |

$PWM_{10}$ score dropped below significance (0.31), and the interaction term turned out to be totally insignificant (*p*-value: 0.97), thus discounting any interaction between the sites in the present dataset. On this basis, the null hypothesis of absence of any interaction could not be rejected, and we concluded that there is little evidence for interaction between the −35 and −10 sites in contributing to promoter strength.

Our model assumed that both the predictors carried independent information about the promoter strength, and together they are able to provide sufficient information about the strength. The basis of this assumption was probed to determine if both predictors are necessary to the model. Could one predictor provide sufficient information about the promoter strength in the absence of the other? There are at least three angles to address this question, and all of them were considered to interpret the model better.

| Table 5 Validation with major (A1, A2, A3) and minor (C, D) promoters. | | | | | |
|---|---|---|---|---|---|
| Promoter | −35 sequence | −10 sequence | Strength | Predicted exp(logStrength) | Predicted class |
| A1 | ttgact | gatact | strong | 0.2904988307 | Medium |
| A2 | ttgaca | taagat | strong | 0.9947607331 | Strong |
| A3 | ttgaca | tacgat | strong | 0.658183377 | Strong |
| C | ttgacg | tagtct | minor | 0.1452865585 | Minor |
| D | ttgact | taggct | minor | 0.1541996302 | Minor |

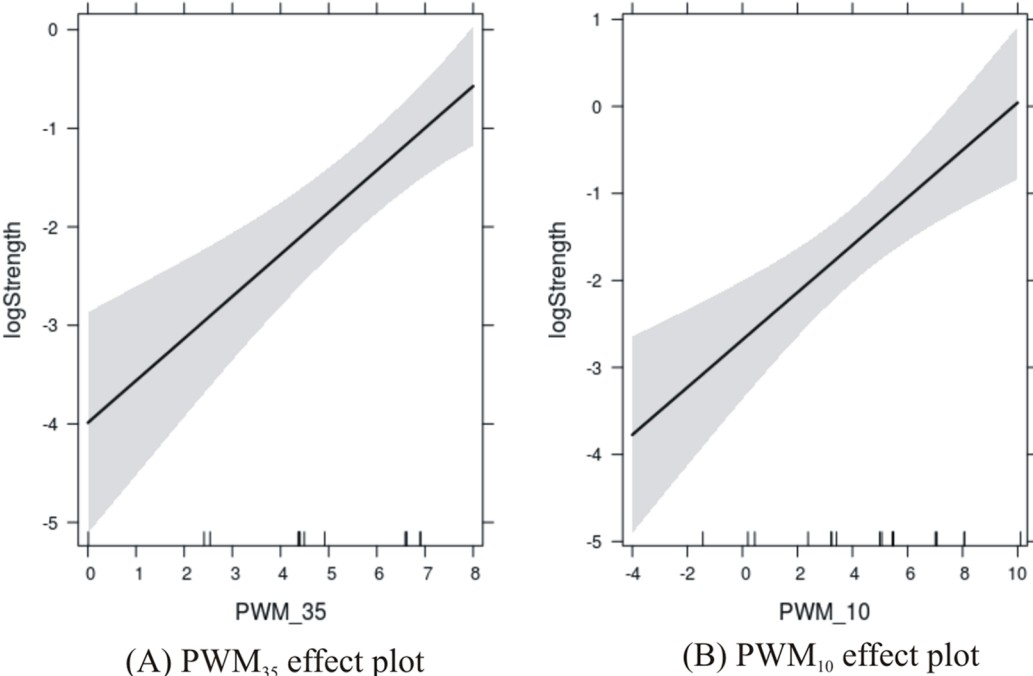

(A) PWM$_{35}$ effect plot          (B) PWM$_{10}$ effect plot

**Figure 3 Effects plots of promoter sites on promoter strength.** (A) −35 promoter site; and (B) −10 promoter site.               

1. Comparing the raw, unadjusted $R^2$ with the adjusted $R^2$. The corresponding values were:

   $R^2 \approx 0.69$

   Adj. $R^2 \approx 0.65$

   Since there is not much difference between $R^2$ and adj. $R^2$, we could say that both predictors contribute substantially to the response variable (promoter strength) and account for about 65% of its variance.

2. Since the $p$-values of both predictors are significant, it would be interesting to observe their effect on the response variable in more detail. This was performed using the `effects` package in R:

```
library(effects)
fit = lm(logStrength~ PWM35+ PWM10, data)
plot(allEffects(fit))
```

The results are shown in Fig. 3 where the PWM scores are plotted against the level of confidence in the predicted response. Confidence in the effect of −35 site increases with the score from 0 to about 7, and then is susceptible to edge effects as the score reaches 8. Confidence in the effect of the −10 site increases with the score from −4 to about 5, and then is susceptible to edge effects as the score reaches 10.

3. Another way to address the question is to compute the correlation coefficients between all the variables of interest, including a variable with the combined effects of −35 and −10 sites. This is shown in Table 6. Three features were used, namely $PWM_{-10}$ score, $PWM_{-35}$ score, and the combined score (i.e. $PWM_{-10}$ + $PWM_{-35}$). These feature variables were correlated with two response variables, namely promoter strength and its corresponding log-transformation. It was first observed that the $PWM_{-10}$ and $PWM_{-35}$ scores were anti-correlated with each other (correlation coefficient = −0.37), thus supporting the hypothesis that they are two independent features that could compensate for each other in determining promoter strength. It was significant that the each feature was better correlated with the log of the strength than the strength itself. We tried to regress the strength on the PWM scores, but the model had a very low adj. $R^2$ (≈0.40) and the intercept term was not significant at the 0.05 level. Further, the highest correlation between the features and response variable was observed between the combined score and log of the promoter strength (~0.79), but the combined score showed only a moderate correlation with the promoter strength prior to log-transformation (~0.63). This was in keeping with similar observations for the strength of $\sigma^E$ promoters (*Rhodius & Mutalik, 2010*) and underscored the logarithmic dependence between the promoter strength and sequence.

Finally, the assumptions of linear modelling were investigated with reference to our problem. Model diagnostics of four basic assumptions were plotted (shown in Fig. 4). Specifically:

Plot A: The residuals were plotted against the fitted values. No trend was visible in the plot, indicating the residuals did not increase with the fitted values and followed a random pattern about zero. This validated the assumption that the errors were independent.

Plot B: The square root of the relative error (standardized residual) was plotted against the fitted value. An almost flat trend was observed, indicating that the standardized residual did not vary with the fitted value. This further validated the assumption that the errors were independent.

Plot C: To test the assumption that the errors were normally distributed, the standardized residuals were plotted against the theoretical quantiles of a normal distribution. The residual distribution closely followed the theoretical quantiles, except for minor deviations towards the tails of the distribution.

Plot D: Since the least-squares cost function is sensitive to outliers, the number of outliers should be kept to a minimum. This was investigated by plotting the standardized residual against the corresponding instance's model leverage. This plot showed that there were no significant outliers in the dataset that could exert an undue influence on the regression parameters.

**Table 6 Correlation matrix of features and response variables.**

| Correlation coefficient | PWM$_{-35}$ | PWM$_{-10}$ | Combined | Strength | Log-strength |
|---|---|---|---|---|---|
| PWM$_{-35}$ | 1 | −0.3715610 | 0.3401672 | 0.4558838 | 0.5153622 |
| PWM$_{-10}$ | −0.3715610 | 1 | 0.7466500 | 0.3025062 | 0.4115533 |
| Combined | 0.3401672 | 0.7466500 | 1 | 0.6330488 | 0.7861173 |
| Strength | 0.4558838 | 0.3025062 | 0.6330488 | 1 | 0.8665495 |
| Log-strength | 0.5153622 | 0.4115533 | 0.7861173 | 0.8665495 | 1 |

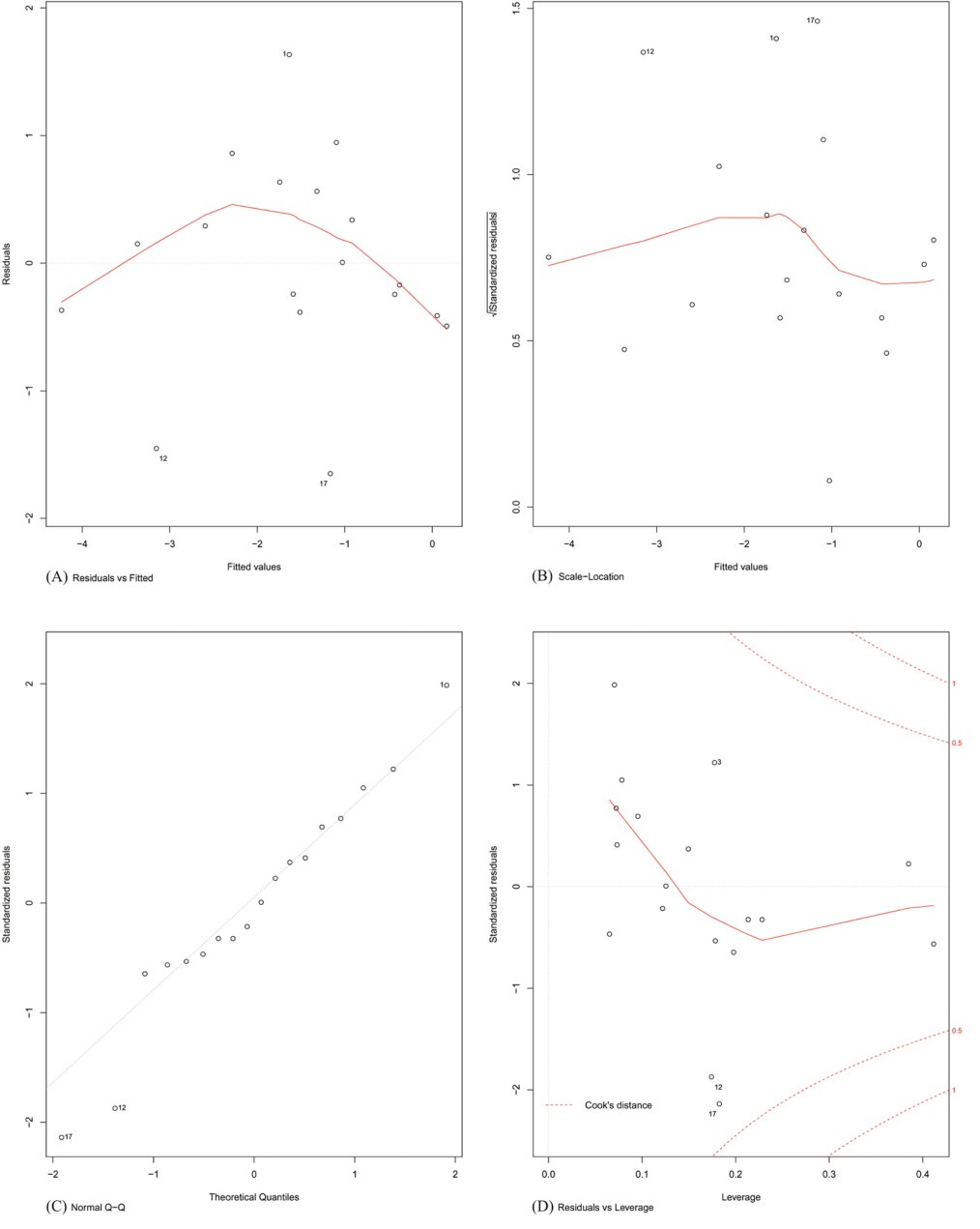

**Figure 4 Model diagnostics plots for investigating the assumptions underlying linear modelling.**
(A) Residuals vs. fitted values; (B) homogeneity of residual variances; (C) normal Q-Q plot; and (D) residuals vs. leverage plot.

An alternative univariate regression model using only the combined score of the PWMs found the coefficient of regression and the F-statistic significant (both $p$-values $\approx 10^{-4}$). However, the adj. $R^2$ of the model ($\approx 0.59$) was much lower than that for Eq. (2), so the original multiple linear regression model was retained for the estimation of the promoter strength.

In summary, our model performed equally well on datasets of strong promoter sequences and datasets of weak random promoter sequences. Our model was consistent in detecting promoter strengths across a 1,000-fold span of promoter strengths in *E. coli* as well as the promoter strengths of a different species, *T. maritima*. The model was further able to discriminate between the major and minor promoters of bacteriophage T7.

Based on these results, an open-access open-source web server and standalone tool offering the prediction service have been implemented. Since the linear modelling results are dependent on the dataset, our implementation provides a facility to augment the learning based on user-provided inputs. The web interface is based on Python web module (web.py) and nginx server. The computational layer is based on numpy, Biopython and matplotlib. The user is provided with an option to add any number of promoter instances with −10 and −35 sequences and the corresponding strengths to augment the training data of the supervised model. The measurement of promoter strength could be done in the manner of *Kelly et al. (2009)*, where the GFP (reporter gene) synthesis rate is measured per unit biomass, and this could be normalized relative to the reference promoter. In order to assess the goodness of fit of the updated model, the $R^2$-value is re-computed, along with the 3D plot of the regression surface. This would enable the user to decide whether the data added to the model has improved its performance for further experiments with the software. Based on the trained model, the user could predict the strength of an uncharacterised promoter given its −10 and −35 hexamers.

## CONCLUSION

The following important conclusions were drawn from our study. (1) Sequence-based modelling yielded a non-linear, logarithmic dependence between promoter strength and sequence. (2) The model was able to discriminate equally well between strong/major promoters and weak/minor/random promoter sequences, indicating successful learning of the essential features of promoter strength prediction. (3) The combined score ($PWM_{-35}$ + $PWM_{-10}$) emerged as the single most important predictor of the promoter strength. Our model yielded robust quantitative prediction across a 1,000-fold span of promoter strengths. It is straightforward to extend our methodology to the study of new promoter classes of other σ factors. Our implementation and web service could be useful in characterizing promoters identified in genome sequencing projects as well in engineering promoters for the design of finely-tuned genetic circuits in synthetic biology. The dynamic feature of our implementation would enable users to incorporate their own data into the model and obtain more reliable estimates of promoter strength. The service will be periodically updated based on the availability of new training instances, user input data and/or models for promoters of other σ factors.

# ACKNOWLEDGEMENTS

We would like to thank the reviewers for helping improve an earlier version of the manuscript. We are grateful for computing facilities at SASTRA Deemed University for support.

### Funding

The authors received no funding for this work.

### Competing Interests

The authors declare that they have no competing interests.

### Author Contributions

- Ramit Bharanikumar performed the experiments, analysed the data, contributed reagents/materials/analysis tools, prepared figures and/or tables, approved the final draft.
- Keshav Aditya R. Premkumar performed the experiments, analysed the data, contributed reagents/materials/analysis tools, prepared figures and/or tables, approved the final draft.
- Ashok Palaniappan conceived and designed the experiments, performed the experiments, analysed the data, contributed reagents/materials/analysis tools, prepared figures and/or tables, authored or reviewed drafts of the paper, approved the final draft.

### Data Availability

Raw code: https://github.com/PromoterPredict/PromoterStrengthPredictor

Web service: https://promoterpredict.com/

Palaniappan, Ashok; Aditya, Keshav; Bharanikumar, Ramit (2018): PromoterPredict: sequence-based modelling of promoter strength: supplementary information. figshare. Fileset. https://doi.org/10.6084/m9.figshare.6794939.v1

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
