# Peer review of "PromoterPredict: sequence-based modelling of Escherichia coli σ70 promoter strength yields logarithmic dependence between promoter strength and sequence"

_PeerJ, doi:10.7717/peerj.5862_

## Round 0.1 · original submission · Major Revisions

The manuscript needs serious update. We had critical remarks demanding rejection. However I believe that this material could be better presented with more specific application. E.coli is a classical model system. As the reviewers suggested the article need more literature references and comparison with existing works. It is worthy to show some novelty, at least to cite recent literature (of year 2018).

·

Basic reporting

Bharanikumar with coauthors developed a regression model to predict the strength of E. coli promoters. Paper is clearly structured and written. Raw data and results of modeling are available

Experimental design

Authors use 13 (actually twelve as promoters BBa_J23103 and BBa_J23112 are the same) to create the linear regression model describing dependency of a logarithm of activity from sequences in -10 and -35 boxes of the promoter. It is known that in average from 12 base pairs in two boxes (-10 and -35 hexanucleotides) only eight is conserved. Together with variation in spacer length, it makes not unusual that one promoter can equally good match several combinations of the boxes and spacer. Authors do not discuss how they choose positions in promoter sequence to calculate position weight matrices (PWMs).
Authors use the approach similar to one provided in Rhodius and Mutalik PNAS,2010. 107, 2854-2859. Rhodius and Mutalik use over 60 promoters in both in vivo and in vitro setup to investigate the correlation of sigmaE promoter strength with PWMs and penalty for non-optimal spacer length. The major concern about the design of computational experiment in the study under review is that the use the same small set of 12 promoters to get PWMs and learn regression coefficients. Together with three regression coefficients, authors learn 48 weights in two PWMs. The number of coefficients to find is higher than the number of samples, which opens the question about the validity of results. It would be interesting to compare your regression results with regression using more general promoter PWMs, obtained in the other studies, for example in Huerta AM, Collado-Vides J (2003) Sigma70 promoters in Escherichia coli: Specific transcription in dense regions of overlapping promoter-like signals. J Mol Biol 333: 261–278.

Validity of the findings

Authors do not test the predictive power of their model, restrict their self with analysis of 12 promoters in training set. However, over about 50 years of investigation of bacterial transcriptions, there are many studies comparing the strength of various promoters in different conditions (to name a few J. Mol. Biol. 1978 125, 467, Plasmin 1996, 35, 108, J. Biol. Chem. 1984, 259, 1616). And despite that direct comparison of those result with each other is rarely possible this data could be used as independent validator set for the regression model provided in the paper. If regression results will be ordered in the same way as the strength value in experiment paper, it proves the validity of the model.
Another aspect missing in the paper is the validation of the model on negative results. It would be interesting to see what would be an average level of strength predicted by the model on non-promoter sequence set.

Additional comments

There are minor remarks on analysis of the regression. The fact that after training the model with combination term demonstrates lost of significance for all components of the model could also be an indicator of overfitting.

·

Basic reporting

Correlation between the promoter sequences and their strength is a matter discussed from the very beginning of the promoter era. The further the more it becomes clear that the use of simple correlations usually leads to erroneous conclusions and any statement needs to be carefully checked. Now, there are many possibilities for this, including a huge amount of expression data that allow sequence-functions comparison for large sets of bacterial promoters. Why not to use them instead of an unrepresentative set of variants of the sequence of only one plasmid promoter.

Experimental design

Very bad: training set is basically unsuitable for predictive modeling.

Validity of the findings

Minor. There are several unexpected observations, but they are suspicious and convincingly not proven.

Additional comments

The authors proposed an approach realized as a freely available resource for predictive modeling of the strength of sigma-70 promoters. This is useful if not misleading. I'm afraid that the authors did not submit any data to prove the adequacy of their approach.

Major concerns:

1. The training set gave consensuses substantially different from those that were created for the entire set of bacterial promoters. Therefore, its suitability for predictive modeling is very doubtful.
2. The training set with 19 samples is too small for decent modeling. It contains identical sequences in both -35 and -10 regions. Thus, the actual number of variables for PWM generation in no way can be considered sufficient, and the authors did not discuss whether and how they took into account this similarity.
3. The predictive ability of the software should be verified on some other data set(s). Several comparative sequence/strength studies were published 25-35 years ago, including exhausting mutagenesis in the same promoter. So, authors are recommended to find these data, starting, for instance, with an article of Knaus and Bujard (1990) ‘Principles Governing the Activity of E. coli Promoters”. In: Eckstein F., Lilley D.M.J. (eds) Nucleic Acids and Molecular Biology, but there are others.
4. At least one of the three main findings of the study is very doubtful: “The –10 and –35 sites were equally important in determining promoter strength”. It directly contradicts the existence of many promoters without the -35 element and must be carefully checked. It is unlikely that it will be confirmed on a representative set of natural promoters. In the second conclusion, that logarithmic scales give a better correlation between scores and strength, on the contrary, there is no novelty. There also an interesting statement that “the PWM—10 and PWM—35 scores were uncorrelated”, which should be validated by the whole set of E.coli s-70 promoters available in RegulonDB.
5. At the end of the paper there is a statement that “The user is provided with an option to add any number of promoter instances with –10 and –35 sequences and the corresponding strengths to augment the training data of the supervised model.” It is unclear how the strength of the new promoters will be integrated into the existing set. At least strategically, the normalization method should be indicated.


Minor concerns:

1. It should be clearly indicated that the training set was composed of plasmid tet promoter variants.
2. All symbols used in the formula: Adj. R2 = 1 – {(1-R2)*[(n-1)/(n-m-1)]} should be deciphered.
3. It is not clear, why Biopython resource was required to convert frequency matrices into log-odds matrices? Too simple procedure to discuss it.
4. Citing: “The two hexamers are located ~10 bp and ~35 bp upstream of the transcription start site, and are called the –10 and –35 sequences respectively (Paget and Helmann, 2003; 41 Kadonaga, 2012). Both -35 and -10 elements were introduced much earlier and must be cited properly.
5. The statement: “Promoters with –10 and –35 sequences matching the consensus motif of the hexamers are typically stronger, meaning they initiate more transcripts per unit time than promoters with less canonical –10 and –35 regions” requires citation as well as mentioning of contradictive data as far as authors want to contribute to this area.

Reviewer 3 ·

Basic reporting

The article proposes a method for predicting the strength of E. coli promoters on the basis of the regression model. The paper is written in professional English. It is structured in accordance with the requirements of the journal and contains raw data.
The article does not have enough literary references and a brief description of the results of other works in which attempts were made to develop computer methods for predicting the strength of prokaryotic promoters (for example, Michael DekhtyarEmail author, Amelie Morin and Vehary Sakanyan Triad pattern algorithm for predicting strong promoter candidates in bacterial genomes BMC Bioinformatics20089:233; Marjan De Mey, Jo Maertens, Gaspard J Lequeux, Wim K Soetaert, and Erick J Vandamme Construction and model-based analysis of a promoter library for E. coli: an indispensable tool for metabolic engineering BMC Biotechnol. 2007; 7: 34.; Joseph H. Davis, Adam J. Rubin, and Robert T. Sauer Design, construction and characterization of a set of insulated bacterial promoters Nucleic Acids Res. 2011 Feb; 39(3): 1131–1141.).

Experimental design

Although now there are non-commercial programs for predicting the strength of prokaryotic promoters, they are not available online (Michael Dekhtyar, Amelie Morin and Vehary Sakanyan Triad pattern algorithm for predicting strong promoter candidates in bacterial genomes BMC Bioinformatics20089:233). The authors offered the first Internet-accessible program to solve this problem.
Based on the regression model, the authors analyzed 19 E. coli promoters and evaluated the contribution of weights of -10 and -35 promoter regions in the final strength of promoter. Unfortunately, the analyzed sample contains a significant number of identical promoter sequences having a similar or identical strength value. This can lead to a significant shift in the estimation of correlation and significance. The use of duplicates is especially dangerous for LOOCV methods, since the ejection of sequences one by one does not release the training set from the control sample.
In addition, Table 3 contains an arithmetical error in the calculation of 1, 5, and 11 rows (-4.6 - - 3.77! = -0.73), which also requires a recalculation of the results.

Validity of the findings

Unfortunately, the authors evaluated the effectiveness of the proposed approach only on sequences from a small training sample containing duplicates. It would be useful to test its effectiveness on other available experimental data (for example, Joseph H. Davis, Adam J. Rubin, and Robert T. Sauer Design, construction and characterization of a set of insulated bacterial promoters Nucleic Acids Res., 2011 Feb; 39 ( 3): 1131-1141.). In spite of the difference in the experimental approaches used, it is possible to obtain a general idea of the reliability of the proposed method on independent data. In addition, on random or non-promoter sequences, it is necessary to look at the distribution of the predicted sequence strength to estimate the error of the method's over-prediction.

---

## Round 0.2 · Minor Revisions

Despite expecting it, we were unable to receive the re-review of one of the reviewers, hence the delay in this decision. However, overall the manuscript has been updated and needs only minor revision. Please consider the remarks from first reviewer.

Reviewer 3 ·

Basic reporting

The authors took into account the recommendations of reviewers and expanded the list of references.

Experimental design

The authors recalculated the data taking into account the detected errors, but Table 1 contains the old values of Predicted ln(Promoter Activity) and must be brought into line with the new results. For example, for hexamers TTGACG and TACAGT, the value -1.46 is indicated in the table, and the program returns -1.63.

Validity of the findings

The authors took into account the wishes of the reviewers and significantly expanded the set of test samples to assess the effectiveness of the proposed approach.

Additional comments

I have no doubt about the usefulness of article submission, but I would recommend to double-check all of the values in the article. In addition, I recommend that you change the settings for the ppp photo on the website, because when you scale the page, it significantly distorts.

---

## Author Rebuttal · Round 0.2

> We would like to thank the Editor and the three reviewers for very insightful comments, which have helped us in substantially improving the manuscript. At the outset, we outline the major changes to our work:
1. We have trained the PWMs on the RegulonDB, and the complete method is outlined in the manuscript
2. We have tested our model on random promoter sequences and found the predictions valid (pls refer the ms.)
3. We have further validated the model on three independent datasets and found the predictions in good order.
Further responses are below and marked by '>'.

Editor's Comments
MAJOR REVISIONS
The manuscript needs serious update. We had critical remarks demanding rejection. However I believe that this material could be better presented with more specific application. E.coli is a classical model system. As the reviewers suggested the article need more literature references and comparison with existing works. It is worthy to show some novelty, at least to cite recent literature (of year 2018).

> We have tried our best and more than trebled the number of references.

Reviewer 1 (Anatoly Sorokin)
Basic reporting
Bharanikumar with coauthors developed a regression model to predict the strength of E. coli promoters. Paper is clearly structured and written. Raw data and results of modeling are available
Experimental design
Authors use 13 (actually twelve as promoters BBa_J23103 and BBa_J23112 are the same) to create
> We have removed BBa_J23103 from the analysis; we now have 18 (not 19) promoters for training the regression model.

the linear regression model describing dependency of a logarithm of activity from sequences in -10 and -35 boxes of the promoter. It is known that in average from 12 base pairs in two boxes (-10 and -35 hexanucleotides) only eight is conserved. Together with variation in spacer length, it makes not unusual that one promoter can equally good match several combinations of the boxes and spacer. Authors do not discuss how they choose positions in promoter sequence to calculate position weight matrices (PWMs).
Authors use the approach similar to one provided in Rhodius and Mutalik PNAS,2010. 107, 2854-2859. Rhodius and Mutalik use over 60 promoters in both in vivo and in vitro setup to investigate the correlation of sigmaE promoter strength with PWMs and penalty for non-optimal spacer length. The major concern about the design of computational experiment in the study under review is that the use the same small set of 12 promoters to get PWMs and learn regression coefficients. Together with three regression coefficients, authors learn 48 weights in two PWMs. The number of coefficients to find is higher than the number of samples, which opens the question about the validity of results. It would be interesting to compare your regression results with regression using more general promoter PWMs, obtained in the other studies, for example in Huerta AM, Collado-Vides J (2003) Sigma70 promoters in Escherichia coli: Specific transcription in dense regions of overlapping promoter-like signals. J Mol Biol 333: 261–278.

> We have used the regulonDB with more than 1000 instances to train the PWMs. This should address the question of effective learning.

Validity of the findings

Authors do not test the predictive power of their model, restrict their self with analysis of 12 promoters in training set. However, over about 50 years of investigation of bacterial transcriptions, there are many studies comparing the strength of various promoters in different conditions (to name a few J. Mol. Biol. 1978 125, 467, Plasmin 1996, 35, 108, J. Biol. Chem. 1984, 259, 1616). And despite that direct comparison of those result with each other is rarely possible this data could be used as independent validator set for the regression model provided in the paper. If regression results will be ordered in the same way as the strength value in experiment paper, it proves the validity of the model.

> Thanks, we have used the suggested references, and made the independent validations.

Another aspect missing in the paper is the validation of the model on negative results. It would be interesting to see what would be an average level of strength predicted by the model on non-promoter sequence set.

> This has also been addressed using a random promoter set.

Comments for the Author
There are minor remarks on analysis of the regression. The fact that after training the model with combination term demonstrates lost of significance for all components of the model could also be an indicator of overfitting.

>This should not be the case after training the PWMs independently using regulonDB
>We would like to thank the reviewer for helpful comments.

Reviewer 2 (Olga Ozoline)
Basic reporting
Correlation between the promoter sequences and their strength is a matter discussed from the very beginning of the promoter era. The further the more it becomes clear that the use of simple correlations usually leads to erroneous conclusions and any statement needs to be carefully checked. Now, there are many possibilities for this, including a huge amount of expression data that allow sequence-functions comparison for large sets of bacterial promoters. Why not to use them instead of an unrepresentative set of variants of the sequence of only one plasmid promoter.
Experimental design
Very bad: training set is basically unsuitable for predictive modeling.

>Hopefully this has been addressed.

Validity of the findings
Minor. There are several unexpected observations, but they are suspicious and convincingly not proven.
Comments for the Author
The authors proposed an approach realized as a freely available resource for predictive modeling of the strength of sigma-70 promoters. This is useful if not misleading. I'm afraid that the authors did not submit any data to prove the adequacy of their approach.

>We have now submitted independent validation  of our approachh. We hope to have made some positive contribution to this area

Major concerns:

1. The training set gave consensuses substantially different from those that were created for the

entire set of bacterial promoters. Therefore, its suitability for predictive modeling is very doubtful.
>Pls see above

2. The training set with 19 samples is too small for decent modeling. It contains identical sequences in both -35 and -10 regions. Thus, the actual number of variables for PWM generation in no way can be considered sufficient, and the authors did not discuss whether and how they took into account this similarity.

>Same as above

3. The predictive ability of the software should be verified on some other data set(s). Several comparative sequence/strength studies were published 25-35 years ago, including exhausting mutagenesis in the same promoter. So, authors are recommended to find these data, starting, for instance, with an article of Knaus and Bujard (1990) 'Principles Governing the Activity of E. coli Promoters". In: Eckstein F., Lilley D.M.J. (eds) Nucleic Acids and Molecular Biology, but there are others.

>Yes, we have independently validated on three other datasets.

4. At least one of the three main findings of the study is very doubtful: "The –10 and –35 sites were equally important in determining promoter strength". It directly contradicts the existence of many promoters without the -35 element and must be carefully checked. It is unlikely that it will be confirmed on a representative set of natural promoters. In the second conclusion, that logarithmic scales give a better correlation between scores and strength, on the contrary, there is no novelty. There also an interesting statement that "the PWM—10 and PWM—35 scores were uncorrelated", which should be validated by the whole set of E.coli s-70 promoters available in RegulonDB.

>Thanks for highlighting this. Indeed with the regulonDb-trained PWMs, the regression coefficients are no longer equal, yet much more significant. So yes, this seems to have been an artifact, not a real observation. We also see now that the PWM scores are anti-correlated.

5. At the end of the paper there is a statement that "The user is provided with an option to add any number of promoter instances with –10 and –35 sequences and the corresponding strengths to augment the training data of the supervised model." It is unclear how the strength of the new promoters will be integrated into the existing set. At least strategically, the normalization method should be indicated.

> We provide the dynamic option to update our legression learner., to expand the scope of our work

Minor concerns:

1. It should be clearly indicated that the training set was composed of plasmid tet promoter variants.
> This has been indicated.

2. All symbols used in the formula: Adj. R2 = 1 − {(1-R2)*[(n-1)/(n-m-1)]} should be deciphered.
> Thanks, this has been done.

3. It is not clear, why Biopython resource was required to convert frequency matrices into log-odds matrices? Too simple procedure to discuss it.

>Thanks, Biopython was used to create the motifs as well, and we used what's existing.

4. Citing: "The two hexamers are located ~10 bp and ~35 bp upstream of the transcription start site, and are called the –10 and –35 sequences respectively (Paget and Helmann, 2003; 41 Kadonaga, 2012). Both -35 and -10 elements were introduced much earlier and must be cited properly.

>We have tried to make all proper citations now. Thank you very much
5. The statement: "Promoters with –10 and –35 sequences matching the consensus motif of the hexamers are typically stronger, meaning they initiate more transcripts per unit time than promoters with less canonical –10 and –35 regions" requires citation as well as mentioning of contradictive data as far as authors want to contribute to this area.

>We would like to thank the reviewer for critical comments.

Reviewer 3 (Anonymous)
Basic reporting
The article proposes a method for predicting the strength of E. coli promoters on the basis of the regression model. The paper is written in professional English. It is structured in accordance with the requirements of the journal and contains raw data.
The article does not have enough literary references and a brief description of the results of other works in which attempts were made to develop computer methods for predicting the strength of prokaryotic promoters (for example, Michael DekhtyarEmail author, Amelie Morin and Vehary Sakanyan Triad pattern algorithm for predicting strong promoter candidates in bacterial genomes BMC Bioinformatics20089:233; Marjan De Mey, Jo Maertens, Gaspard J Lequeux, Wim K Soetaert, and Erick J Vandamme Construction and model-based analysis of a promoter library for E. coli: an indispensable tool for metabolic engineering BMC Biotechnol. 2007; 7: 34.; Joseph H. Davis, Adam J. Rubin, and Robert T. Sauer Design, construction and characterization of a set of insulated bacterial promoters Nucleic Acids Res. 2011 Feb; 39(3): 1131–1141.).
Experimental design
Although now there are non-commercial programs for predicting the strength of prokaryotic promoters, they are not available online (Michael Dekhtyar, Amelie Morin and Vehary Sakanyan Triad pattern algorithm for predicting strong promoter candidates in bacterial genomes BMC Bioinformatics20089:233). The authors offered the first Internet-accessible program to solve this problem.

>Thank you. Dekhtyar et al provide a very limited use ("strong promoter" prediction) in the context of MS-Word. We provide a cross-platform Python implementation that could be easily adapted for other uses, plus a web-server.

Based on the regression model, the authors analyzed 19 E. coli promoters and evaluated the contribution of weights of -10 and -35 promoter regions in the final strength of promoter. Unfortunately, the analyzed sample contains a significant number of identical promoter sequences having a similar or identical strength value. This can lead to a significant shift in the estimation of correlation and significance. The use of duplicates is especially dangerous for LOOCV methods, since the ejection of sequences one by one does not release the training set from the control sample. In addition, Table 3 contains an arithmetical error in the calculation of 1, 5, and 11 rows (-4.6 - -3.77! = -0.73), which also requires a recalculation of the results.

>The entire modelling has been done. We are sorry for our oversight.

Validity of the findings
Unfortunately, the authors evaluated the effectiveness of the proposed approach only on sequences from a small training sample containing duplicates. It would be useful to test its effectiveness on other available experimental data (for example, Joseph H. Davis, Adam J. Rubin, and Robert T.

Sauer Design, construction and characterization of a set of insulated bacterial promoters Nucleic Acids Res., 2011 Feb; 39 ( 3): 1131-1141.). In spite of the difference in the experimental approaches used, it is possible to obtain a general idea of the reliability of the proposed method on independent data. In addition, on random or non-promoter sequences, it is necessary to look at the distribution of the predicted sequence strength to estimate the error of the method's over-prediction.

>Both the independent validation and testing on a random promoter set have been done and included in the manuscript.

>We would like to thank the reviewer for constructive insightful comments.

---

## Round 0.3 · accepted · Accept

The reviewers have no more remarks.

# ·

Basic reporting

Authors have improved manuscript and take into account all concerns of reviewers.

Experimental design

Authors incorporate both negative set analysis and additional promoter strength data.

Validity of the findings

Results are now statistically sound.

Reviewer 3 ·

Basic reporting

The authors took into account all the comments of reviewers and the article is ready for publication.

Experimental design

The authors took into account all the comments of reviewers and the article is ready for publication.

Validity of the findings

The authors took into account all the comments of reviewers and the article is ready for publication.